# Numerical Investigation on the Heat and Mass Transfer in Microchannel with Discrete Heat Sources Considering the Soret and Dufour Effects

**DOI:** 10.3390/mi13111848

**Published:** 2022-10-28

**Authors:** Xueyu Ou, Ruijin Wang, Tongwei Guo, Chun Shao, Zefei Zhu

**Affiliations:** School of Mechanical Engineering, Hangzhou Dianzi University, Hangzhou 310018, China

**Keywords:** heat and mass transfer, Soret effect, discrete heat sources, Dufour effect, micro-channel heat sink

## Abstract

Heat-transfer enhancement in microchannel heat sinks (MCHS) has been a hot topic in the last decade. However, most published works did not focus on the heat sources that are discrete, as in most microelectronic devices, and the enhancement of heat and mass transfer (HMT) due to the Soret and Dufour effects being ignored. Based on a heterogeneous two-phase model that takes into consideration the Soret and Dufour effects, numerical simulations have been performed for various geometries and heat sources. The numerical results demonstrate that the vortices induced by a heat source(s) can enhance the heat transfer efficiency up to 2665 W/m^2^·K from 2618 W/m^2^·K for a discrete heat source with a heat flux *q* = 10^6^ W/m^2^. The Soret effect can affect the heat transfer much more than the Duffour effect. The integrated results for heat transfer due to the Soret and Dufour effects are not sampled superpositions. Discrete heat sources (DHS) arranged in microchannels can enhance heat transfer, especially when the inlet velocity of the forced flow is less than 0.01 m/s. This can provide a beneficial reference for the design of MCHS with DHS.

## 1. Introduction

Convective mixing and heat transfer are very common in microfluidic systems. Nanofluid, as a working fluid, is often employed to enhance heat transfer in MCHS and photovoltaic systems [1,2,3]. Akhter et al. [4] numerically simulated the convective heat transfer of Fe_3_O_4_-water nanofluids in a fan-shaped annulus cavity, and conducted the analysis on the effects of Rayleigh number, cavity structure and volume fraction of nanofluid on the heat transfer. Du et al. [5] investigated the heat transfer enhancement of a MCHS in the presence of an external magnetic field, based on the Buongiorno model when the applied magnetic field and thermophorestic effect are in consideration. Wang et al. [6] indicated that electrophoresis can significantly influence the heat transfer of nanofluid when external electric field is applied. Kumar et al. [7] found that the heat transfer rate in ternary hybrid nanofluid increases by about 2–5%; whereas, in Nanolubricant, it is about 3–8% for the gradual increase of values of the ferromagnetic interaction parameter. Alizadeh et al. [8] studied the convective heat transfer in a cavity, and made a comparison of the enhancement in heat transfer for different heat source patterns. Rostamzadeh et al. [9,10] discussed the influence of hydrophobic and hydrophilic surfaces on heat and mass transfer.

However, HMT induced by the Soret effect and the Dufour effect is generally neglected in the above investigations because they have a much smaller influence on the HMT than those predicted by Fourier’s law. In the presence of temperature and concentration gradients in MCHS, the Soret and Dufour effects may have a considerable influence on HMT. The Soret effect is the mass flux induced by a temperature gradient, and the Dufour effect is the energy flux contributing to a mass concentration gradient. They are reciprocal phenomena of each other. Hayat et al. [11] studied MHD flow and believed that the Soret and Dufour variables can be utilized to regulate the heat transfer intensity. Hou et al. [12] investigated the HMT over a stretched porous surface by considering the Soret and Dufour effects based on a pseudo-plastic model. Wang et al. [13] established a specialized model to simulate the double-diffusion convection in a cavity under different Rayleigh numbers, Dufour numbers and Soret numbers. For larger Dufour and Soret effects, a larger enhancement in HMT can be achieved, due to the induced oscillatory flow. Kumar et al. [14] believed that the mass transfer improves with the Soret number, and the heat transfer increases with the Biot number. Yadav et al. [15] studied the onset of MHD convection and found that a greater diffusion can be achieved, due to the Soret effect, when the temperature on the upper wall is lower. Rghif et al. [16] analyzed the heat transfer in a Solar Pond when the Dufour and Soret effects were taken into consideration. The results demonstrated that the thermosolutal convection, due to the Dufour effect, is more applicable than that owing to the Soret effect, since a larger Dufour number leads to a greater heat transfer. Srinivasacharya et al. [17] analyzed the influence of the Soret number and Dufour number on the flow pasting a wavy surface. Salleh et al. [18] found that the efficiency of HMT increased by 40.43% and 31.41% when the Dufour and Soret effects were taken into account, respectively. Jawad et al. [19] simulated the MHD laminar flow and analyzed the influences of the Soret number, Schmidt number, Dufour number and Prandtl number on the heat transfer.

It is regretful that all of the aforementioned works do not consider the Brownian motion and thermophoresis of the contained nanoparticles. Mondal et al. [20] estimated the convective heat transfer of MHD when the heterogeneous heat source and the thermophoretic effect are taken into account, as well as the Dufour and Soret effects. In addition, several published works have focused on the effect of the Dufour and Soret effects on heat transfer, and the effects of thermophoresis and Brownian motion of contained nanoparticles have also been considered. For instance, Sardar et al. [21] analyzed the HMT of nanofluid under various Prandtl numbers, Soret numbers and Dufour numbers. Raju et al. [22] studied the 2D unsteady natural convection when a transverse magnetic field is applied. It was found that a larger Soret number leads to a larger enhancement in heat transfer and a smaller Dufour number leads to a larger enhancement in mass transfer. Akram et al. [23] investigated the double-diffusive convection induced by the external magnetic field in four tapered asymmetric channels. It is shown that the temperature rises with the Brownian coefficient, Soret number, Dufour number and thermodynamic parameters. Mosayebidorcheh et al. [24] researched the asymmetric peristaltic flow of nanofluid and found consistent conclusions as that in Reference [23]. Tripathi et al. [25] conducted numerical simulations to insight into the double-diffusive convection considering both magnetic field and thermal radiation.

DHS are quite common in practical engineering. Bahiraei and Mashaei [26] investigated the HMT of nanofluid in a channel with DHS. Mebarek-Oudina et al. [27] performed numerical investigations on the MHD natural convection of magnetized nanofluid in an annulus. Hussain et al. [28] evaluated the influence of inclination angle on heat-transfer enhancement inside a partially heated cavity. However, the effect of the Dufour and Soret effect on HMT is not taken into account in the works mentioned above. Of course, there are works in which the Dufour and Soret effects are also considered. For example, Mebarek-Oudina and Bessaïh [29] investigated the natural convection induced by Soret and Dufour effects when two DHSs were arranged in an annulus enclosure. Parveen and Mahapatra [30] studied 2D MHD convection inside a partially heated cavity. Mondal and Mahapatra [31] investigated the mixed convection of magnetic nanofluid in a trapezoidal enclosure with DHSs. It is a pity that all these works concern only the natural convection in an enclosure or cavity.

Overall, most of the above-published works have shown that the Dufour and Soret effects cannot be neglected in the case of large temperature and concentration gradients, since chaotic convection is easily induced by the Dufour and Soret effects. The effects of thermophoresis and Brownian motion of nanoparticles have a distinct effect on HMT, since thermophoresis induced by temperature gradients can carry heat from higher to lower temperatures. Until now, very few studies on discrete heat sources have considered the Soret and Dufour effects in conjunction with Brownian motion and thermophoresis of nanoparticles. This is the first exploration of heat transfer in MCHS with DHS considering both the Soret effect and the Dufour effect. The underlying mechanisms will be discussed also.

## 2. Mathematical Model

### 2.1. Control Equations

Qu et al. [32] found that momentum equations and energy equations can accurately predict the HMT characteristics in microchannels under the condition that the characterized dimension of the channel is larger than 100 mm. In order to take the thermophoresis and Brownian motion of nanoparticles, as well as the Soret effect and the Dufour effect into account, a 2D mathematical model based on the Buongiorno model [33] is established. For unstable flows, the mass, momentum, energy and species equations for the fluid model with the Soret and Dufour effects are as follows [24]:(1)$$\frac{\partial \rho_{nf}}{\partial t}+\nabla \cdot \rho_{nf}\vec{V}=0$$
(2)$$\rho_{nf}\left[\frac{\partial \vec{V}}{\partial t}+\left(\vec{V}\cdot \nabla \right)\vec{V}\right]=-\nabla p+\mu_{nf}\nabla^{2}\vec{V}+\rho_{nf}\vec{g}$$
(3)$$\rho_{nf}\left[\frac{\partial T}{\partial t}+\vec{V}\cdot \nabla T\right]=\frac{k_{f}}{\rho_{f}c_{f}}\nabla T+\rho_{p}c_{p}\left[D_{B}\nabla \varphi \cdot \nabla T+D_{T}\frac{\nabla T\cdot \nabla T}{T}\right]+\frac{D_{m}K_{T}}{c_{s}c_{p}}\nabla^{2}\varphi$$
(4)$$\frac{\partial \varphi }{\partial t}+\vec{V}\cdot \nabla \varphi =\nabla \cdot \left[D_{B}\nabla \varphi +D_{T}\frac{\nabla T}{T}\right]+\frac{D_{m}K_{T}}{T_{m}}\nabla^{2}T$$
where $\vec{V}$ is the velocity, ρ,μ are the density and viscosity, p, *g* are pressure and gravity, respectively. Note, the first term on the right hand of Equation (3) indicates the conductive heat, the second term takes the heat transported by thermophoresis and Brownian motion of nanoparticles into account, the third term considers the Soret effect. The first and second terms on the right hand of Equation (4) are diffusion induced by thermophoresis and the Brownian motion and the Dufour effect. Where the Brownian coefficient and thermophoretic diffusion coefficient are $D_{B}=\frac{K_{B}T}{3\pi \mu_{f}d_{p}}$ and $D_{T}=\beta \frac{\mu_{f}}{\rho_{f}}\varphi =0.26\frac{k_{f}}{2k_{f}+k_{p}}\frac{\mu_{f}}{\rho_{f}}\varphi$, respectively. T,c,φ,k are temperature, specific heat, mass fraction of nanoparticles and thermal conductivity, respectively. $D_{m},\;K_{T},\;K_{B}$ are mass diffusion coefficient, thermal diffusion ratio and Boltzman constant, respectively. $c_{s}$ indicates the concentration sensitivity. The subscripts *p*, *f* and *nf* represent the nanoparticle, base fluid and nanofluid, respectively.

### 2.2. Physical Properties

Al_2_O_3_-water nanofluid is the most commonly used fluid in engineering due to its outstanding thermal conductivity. The physical parameters of the fluid, density, specific heat, thermal diffusivity and viscosity, are related to the physical properties of the composition according to Maxwell’s effective medium theory. The physical properties of additive Al_2_O_3_ and based fluid water can be used to estimate the physical properties of nanofluids (Table 1).

It is known that the dynamic viscosity of water will change with the temperature [6]:(5)$$\mu_{f}=(2.414\times 10^{-5})\times 10^{\frac{247.8}{\left(T-140\right)}}$$

The dynamic viscosity of nanofluid can be written based on the Brinkman model as [34]:(6)$$\mu_{nf}=(1+2.5\varphi )\mu_{f}$$

The density, heat specific, thermal diffusivity and thermal expansivity of nanofluid read as [35]:(7)$$\rho_{nf}=\varphi \rho_{p}+(1-\varphi )\rho_{f}$$
(8)$${\left(c\rho \right)}_{nf}=\varphi {\left(c\rho \right)}_{p}+(1-\varphi ){\left(c\rho \right)}_{f}$$
(9)$$\alpha_{nf}=k_{nf}/{\left(\rho c_{p}\right)}_{nf}$$
(10)$${\left(\rho \beta \right)}_{nf}=\varphi {\left(\rho \beta \right)}_{p}+\left(1-\varphi \right){\left(\rho \beta \right)}_{f}$$

The thermal conductivity of Al_2_O_3_-water nanofluid can be written as [36]:(11)knfkf=1+64.7φ0.75(dfdp)0.37(kpkf)0.75Pr Re1.23
where Re and Pr are Reynolds number and Prandtl number, respectively. It is necessary to define parameters. Rayleigh number to assess the heat transfer performance of natural convection can be read as:(12)$$Ra=\frac{g\beta_{nf}\left(T_{w}-T_{f}\right)L^{3}}{\alpha_{nf}\mu_{\mathrm{nf}}}$$
where *L* is the characteristic dimension, *g* is gravitational acceleration, $T_{w}$ is wall temperature, $T_{f}$ indicates the average temperature of the fluid at the fixed position. The heat transfer coefficient h is:(13)$$h=\frac{q}{T_{w}-T_{f}}$$
where q is heat flux. In some studies, Nusselt’s number $Nu=hL/k_{nf}$ is used to evaluate the convective heat transfer. For comprehensive evaluation of the heat transfer and pressure drop, Nusselt’s number at identical pump power or PEC can be employed [6,37].

### 2.3. Validation of Numerical Model

To validate the above model, numerical simulations of HMT in a 2D fan-shaped ring as shown in Figure 1 were performed. Al_2_O_3_ nanoparticles with a size of 20 nm and 5 wt%. The temperature of the heat source is set to 350 K. The domains occupied by both water and nanofluid are meshed, the number of grids is 6000, 12,000, 18,000 and 20,000, respectively. The corresponding Nusselt’s numbers differ by 2.1%, 0.71% and 0.69%, respectively. This signifies that the grid size of 12,000 is fine enough for the present simulations. In addition, appropriate CPU times of 45 min and 325 min correspond to 6000 and 20,000 grids. The simulations were performed in Fluent 17.0 with finite volume methods for discretization. The Soret and Dufour effects, can be taken into account by User Defined Functions (UDF), programmed in C language. A switch is set to call the UDFs for the Soret or Dufour effects so that the calculations for the individual Soret effects and Dufour effects, and both effects can be performed. Data collections are performed in each iteration, the variables are velocities, temperatures and mass fractions of nanofluid. Comparisons in Table 2 for different Rayleigh numbers with that in Reference [38] showed good agreement, the errors are less than 6.7%. This proves that the above model based on the Buongiorno model is valid.

## 3. Results and Discussions

### 3.1. The Flows in a Fan-Shaped Ring

To explore what kind of flow can be induced by a heat source in a fan-shaped ring, simulations with only one heat source at the top wall were performed to obtain the distribution of nanoparticles and temperature, as well as the velocity field. The parameters and geometry are the same as in Section 2.3. Figure 2a demonstrated the variation of the A1_2_O_3_ mass fraction of reference points 1 and 2 with time. The fact that the concentrations at reference points 1 and 2 tend to be equal means that the nanofluid and water mix uniformly, since reference points 1 and 2 are the most difficult places for total mixing. This can be explained by Figure 2b, which shows the concentration of nanoparticles in the fan-shaped ring at various times. The flow in the fan-shaped ring is depicted in the second row of Figure 2a in the case of one heat source is arranged at the middle top of the top wall. A pair vortex can be seen in the fan-shaped ring. Why can such a vortex pair exist can be interpreted by the distribution of the temperature and concentration in the fan-shaped ring (see Figure 2a). First, the mechanisms of mass transfer mainly include diffusion due to concentration gradients and Brownian motion, thermophoresis of nanoparticles due to temperature gradients, and diffusion due to Soret effects. Second, the mechanisms of heat transfer involve conductive heat due to temperature gradients, diffusive heat due to thermophoresis and Brownian diffusion of nanoparticles, and heat flux induced by the Dufour effect.

Moreover, HMT is different for the different cases with 1, 2 and 3 heat sources. Figure 3 shows us the arrangement of the heat sources, the temperature and flow, and the concentration of the nanoparticles. It can be seen that the mass fractions of the nanoparticles are not very different, while the temperature is higher when more heat sources are arranged. A pair vortex is seen in all three cases, although the largest velocities are induced when only one heat source is arranged. The reason is that the flows induced by different heat sources may interfere with each other. It is therefore crucial to explore which factors can lead to a larger effect on HMT. This will be discussed in a forthcoming subsection.

### 3.2. Influence of the Dufour and Soret Effects

First, the mass fraction of Al_2_O_3_ at reference point 1 is analyzed for various cases, neither the Soret nor the Dufour effect, only the Soret effect, only the Dufour effect, and both the Soret and Dufour effects are considered. A 350 K heat source is arranged at the bottom-middle wall of the fan-shaped ring with a nanoparticle mass fraction of 5 wt%. Figure 4 shows that the Soret effect can produce a larger mass transfer than the Dufour effect. However, the effects of the Dufour and Soret effects on HMT are not simply additive, but combined and interfere. It is easy to understand that a larger mass transfer can be obtained when both the Soret and Dufour effects are considered than when neither the Soret nor the Dufour effects are considered.

It is worth noting that the concentrations in Figure 4 for reference point 1 affected by the Soret effect are higher than that by both the Soret and Dufour effect when t < 6.6 s. Reversely, the concentration induced by the Soret effect is lower than that by both the Soret and Dufour effect when t > 6.6 s. The reason is that, during the first phase of the sharp decrease in the mass fraction at reference point 1, the presence of the temperature and concentration gradient causes the velocity in the flow field to increase and the concentration at reference point 1 to decrease. Figure 5 shows the flow field at t = 3 s for various cases. Concentration gradients promote the diffusion of nanoparticles at high concentrations, but the velocity decreases as the concentration gradient continues to decrease. Hence, in the absence of the Dufour effect and the Soret effect, the concentration gradient in the flow field will be minimal and its velocity will be, instead, minimal. Conversely, the concentration gradient is the largest and its velocity is the largest under the influence of the Soret effect.

The temperature gradient gradually decreases and goes to zero with time. At t = 10 s, the order of the concentration gradient is the case for neither the Soret nor the Dufour effect < only the Dufour effect < only the Soret effect < both the Soret and Dufour effects. Hence, for both the considered Soret and Dufour effects, the velocity in the flow domain is highest. However, the velocity of the flow domain is only slightly smaller for the sole Soret effect than for both the Soret and Dufour effects considered. That is, the effect on the flow, which is affected by both the Soret and Dufour effects, is not a simple superposition of the Soret and Dufour effects. The influences of the various cases on mass transfer are consistent at the initial stage. However, when the temperature and concentration gradients are small, the mutual influence of concentration and temperature on the velocity is somewhat larger. This is because the nanoparticle migrates more rapidly to the location of reference point 1, and the concentration at reference point 1 is accordingly enhanced.

In order to see the heat transfer in such a geometry, the heat transfer coefficients for various cases and at various times were calculated and listed in Table 3. Similar results as that for mass transfer can be drawn. The heat transfer coefficients for various cases are sorted as: neither the Soret effect nor the Dufour effect < only the Dufour effect < only the Soret effect < both the Soret and Dufour effects are considered. The mechanism is induced for convection, in the same way as for mass transfer.

### 3.3. Heat Transfer in Microchannel with Discrete Heat Sources

It is known from Section 3.2 that the DHSs can intensify the convective heat transfer owing to the induced pair vortex. The aim of this subsection is to investigate the HMT in a straight microchannel with DHSs. Bazylak et al. [39] believed that thermal performance is related to the distribution of discrete heat sources because a reasonable flow structure can be formed when the size and space of discrete heat sources are equal. Therefore, the heat transfer in a microchannel with five discrete heat sources (Figure 6a) was numerically simulated in the present work. The dimensions are set to be: the length of microchannel L1 = 4.4 mm, the width of channel W = 0.4 mm, the length of heat source and space of discrete heat source L2 = L3 = 0.4 mm. The mass fraction of nanofluid is 4 wt%, the heat flux of heat sources *q* = 2 × 10^5^, 5 × 10^5^ and 10^6^ W/m^2^, the wall temperature and fluid temperature are set to be 300 K. The physical parameters of the nanofluid and water are the same as that in Table 1. Natural convective heat transfer in microchannels was first studied. The temperature and flow in the microchannel at t = 1.0 s are shown in Figure 6b,c. Five pair vortices corresponding to the discrete heat sources in the microchannel can be clearly seen. Obviously, these pair vortices can enhance the heat transfer owing to the induced chaotic convection. The heat-transfer coefficients for various Rayleigh numbers were calculated and depicted in Figure 7. The heat-transfer coefficient is found to increase with the Rayleigh number and mass fraction.

Then, the forced convective heat transfer for a variety of inlet velocities was studied. Figure 8 shows the temperature distributions together with velocity vectors for *q* = 5 × 10^5^ W/m^2^ when the inlet velocities are 0.0, 0.001, 0.01, 0.1 and 1.0 m/s. For definiteness, the temperature profile in Figure 8 for zero inlet velocity is at t = 1 s, and the other profiles for inlet velocities 0.001, 0.01, 0.1 and 1.0 m/s are at steady state. Figure 8 demonstrates that there is a clear velocity perturbation at inflow velocities smaller than 0.001 m/s, since the maximum velocity in the fluid domain is of the order of 10^−3^ m/s and is nearly independent of the inlet velocity. Reversely, the maximum velocity is always related to the inlet velocity when the inlet velocity is greater than 0.01 m/s. In addition, the most important index, the maximum temperature on the wall where the heat source is located, is 370, 357, 352, 339, 313 and 304 K, respectively, at an inlet velocity of 0.0, 0.0001, 0.001, 0.01, 0.1 and 1.0 m/s. The highest temperatures on the wall with the heat source are listed in Table 4 for various *q* = 10^6^, 5 × 10^5^, 2 × 10^5^ W/m^2^, and it is found that the inlet velocity v_in_ = 0.01 m/s is suitable for all three cases with *q* = 10^6^, 5 × 10^5^, 2 × 10^5^ W/m^2^ if the highest temperatures on the wall with heat source have to be lower than 353 K (80 °C).

In addition, the comparison of the heat transfer coefficients for discrete heat sources with a heat flux *q* = 10^6^ W/m^2^ and that for uniform heat sources with a heat flux *q* = 5 × 10^5^ W/m^2^ is depicted in Figure 9. It is found that the heat-transfer coefficient for a discrete heat source is significantly higher (47 W/m^2^·K) than that for a uniform heat source because a series of pair vortices can enhance the heat transfer when *v*_in_ ≤ 0.01 m/s. Unlike the heat-transfer coefficient for the case with a uniform heat source, it rises first and then falls when 0 ≤ *v*_in_ ≤ 0.004 m/s, and keeps rising when *v*_in_ ≥ 0.004 m/s for the case with a discrete heat source, because there exists obvious flow disturbance induced by the pair vortices.

## 4. Conclusions

Numerical simulations have been performed on fan-shaped rings and rectangular microchannels with discrete heat sources based on a modified Buongiorno model that takes into consideration the Dufour and Soret effects. The effects on HMT are then analyzed for natural and forced convection by considering the individual Soret effect, the individual Dufour effect and both the Dufour and Soret effects together. The main findings are:(1)Based on the Buongiorno model, a modified model was proposed by adding a term for the Soret effect in the momentum equation and another term for the Dufour effect in the energy equation. Such models are suitable for modeling HMT in MCHS or heat exchangers with two species or discrete heat sources.(2)The Soret effect influences HMT more greatly than the Dufour effect. The result, due to the combined the Dufour and Soret effects, is better than the result due to the individual Soret or Dufour effects, but it is not simply due to the superposition of two individual factors, as they will interact with each other.(3)The heat-transfer coefficient for the discrete heat source is substantially greater than that for the uniform heat source when the inlet velocity <0.01 m/s. It first rises and then falls for inlet velocities ranging from zero to 0.004 m/s, and continues to rise for inlet velocities larger than 0.004 m/s due to the presence of visible flow perturbations.(4)The underlying mechanism is that severer convection can be provoked when the Dufour effect and Soret effect are considered, because a pair vortex can be generated near each DHS. This can provide beneficial guidance for the design of MCHS with DHS in the future.

## Figures and Tables

**Figure 1 micromachines-13-01848-f001:**
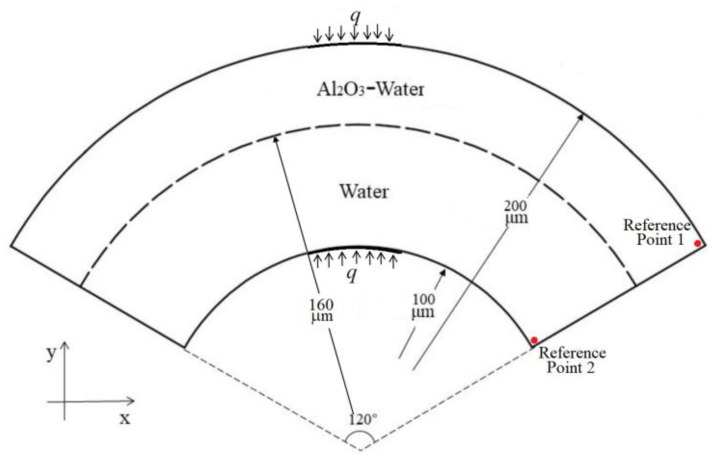
The physical model for validation of numerical model.

**Figure 2 micromachines-13-01848-f002:**
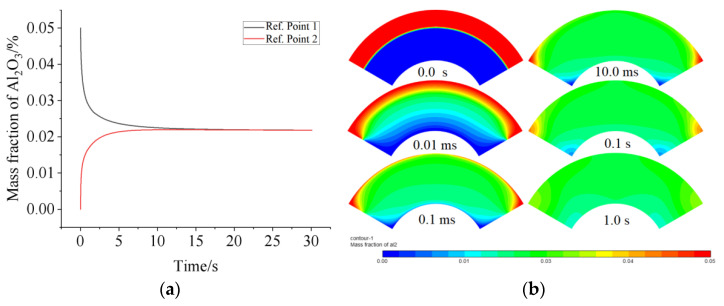
The mass fraction of reference point 1 and point 2 (**a**), and the diffusion process is evaluated by the concentration of nanoparticles at various times. The initial mass fraction of Al_2_O_3_ is 5 wt% (**b**).

**Figure 3 micromachines-13-01848-f003:**
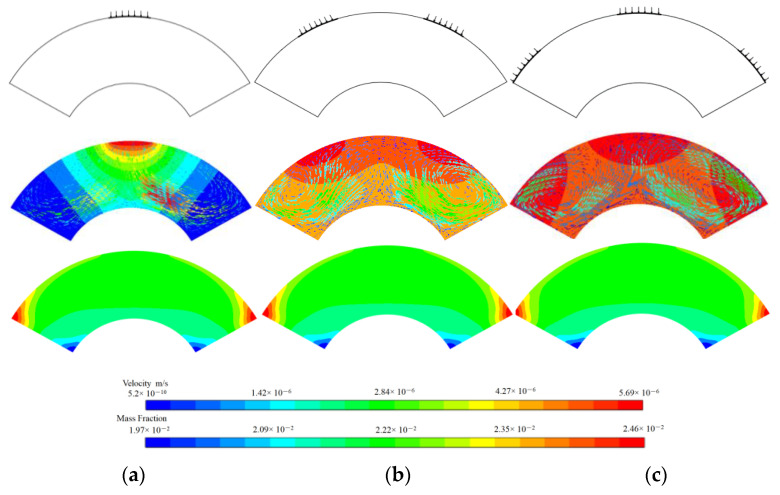
The heat sources (first row), temperatures and flows (second row), mass fraction fields (third row) 3 s after the heat sources were loaded. (**a**) one heat source on the middle top, (**b**) two symmetry heat sources on the top, (**c**) three heat sources on the top. The temperature of heat sources is 350 K, the initial mass fraction of nanoparticle is 5 wt%.

**Figure 4 micromachines-13-01848-f004:**
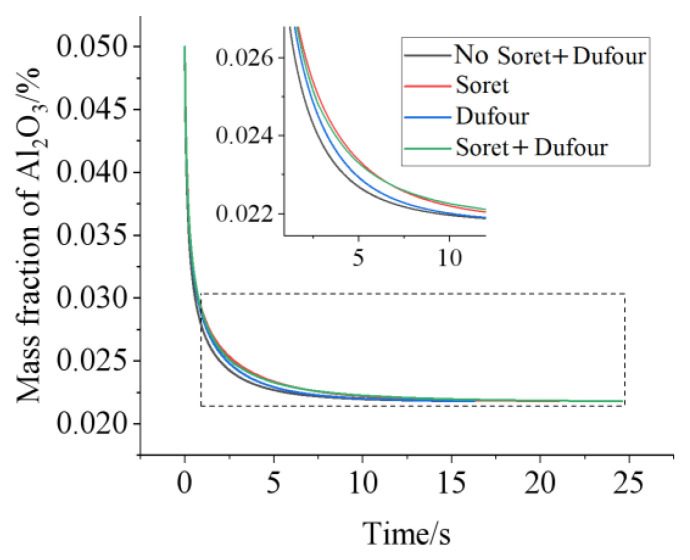
Comparisons of mass transfer for various cases, neither Soret effect nor Dufour effect, only the Soret effect, only the Dufour effect, both Dufour and Soret effects being in consideration. The temperature of heat sources is 350 K, the initial mass fraction of nanoparticle is 5 wt%.

**Figure 5 micromachines-13-01848-f005:**
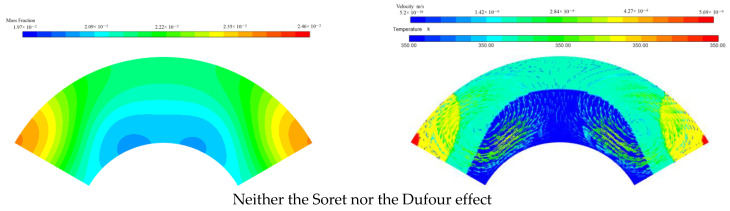
Comparisons of concentrations (left column) and velocities with temperatures (right column) at t = 3 s for various cases, neither Soret effect nor Dufour effect, only the Soret effect, only the Dufour effect, both Soret and Dufour effects being in consideration. The temperature of heat sources is 350 K, the initial mass fraction of nanoparticle is 5 wt%.

**Figure 6 micromachines-13-01848-f006:**
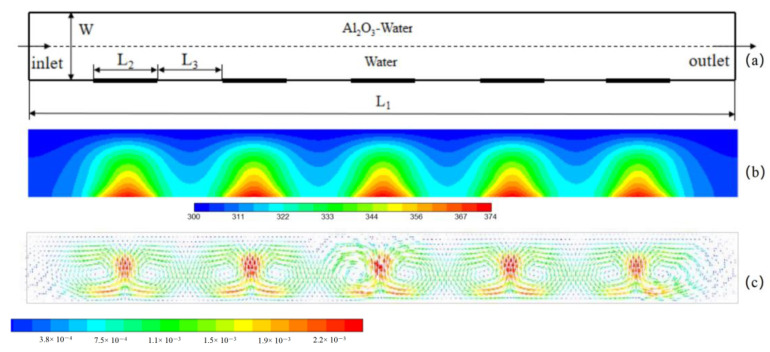
The geometry of microchannel with discrete heat sources (**a**), the temperature distribution in microchannel (**b**) and the flow velocity induced by heat sources (**c**) for *q* = 10^6^ W/m^2^, initial mass fraction 4.0 wt%, initial temperature of fluids and walls 300 K.

**Figure 7 micromachines-13-01848-f007:**
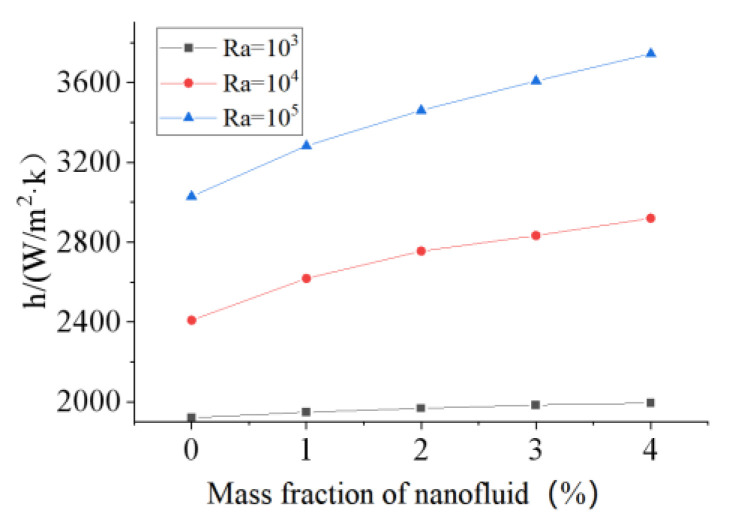
The heat transfer coefficients for various Rayleigh numbers 10^3^, 10^4^ and 10^5^, and mass fractions 0–4 wt%.

**Figure 8 micromachines-13-01848-f008:**
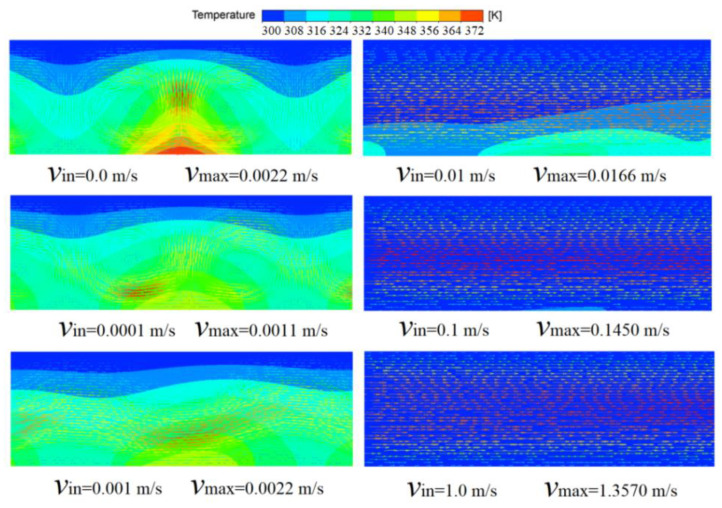
The temperatures with the velocity vector for various inlet velocity when *q* = 5 × 10^5^ W/m^2^.

**Figure 9 micromachines-13-01848-f009:**
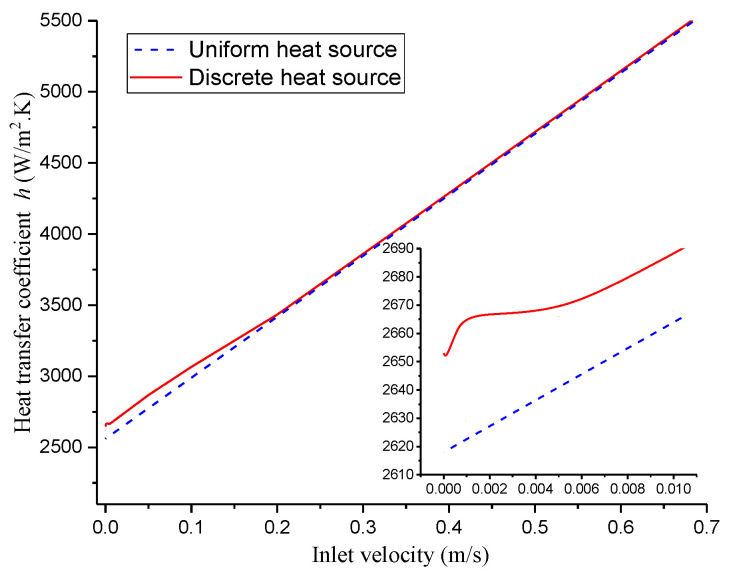
The variation of heat transfer coefficient with the inlet velocity for different heat sources, discrete and uniform heat sources.

**Table 1 micromachines-13-01848-t001:** The physical properties of Al_2_O_3_ and water.

Parameters	$\rho \left(kg/m^{3}\right)$	k(W/m×K)	$c_{p}\left(J/kg\times K\right)$	$\alpha \times 10^{7}\left(m^{2}/s\right)$	$\beta \times 10^{6}\left(1/K\right)$
Water	998.2	0.6	4182	1.47	210
Al_2_O_3_	3970	45	880	131.7	8.5

**Table 2 micromachines-13-01848-t002:** Comparisons of the numerical results and that presented in Reference [38].

*Ra*	Present	Reference [38]	Error
10^5^	4.689	4.465	4.8%
10^6^	9.437	8.838	6.3%

**Table 3 micromachines-13-01848-t003:** The heat transfer coefficients for various cases and at various times (W/m^2^·K).

Cases	0.1 s	1 s	2 s	4 s	6 s
Neither Soret Nor Dufour	5944.0	3784.8	3716.9	3718.0	3718.2
Only the Soret	5956.6	3785.6	3741.0	3739.2	3738.8
Only the Dufour	5947.9	3767.1	3721.7	3719.7	3719.5
Both Soret and Dufour	5966.3	3797.8	3752.1	3749.9	3749.5

**Table 4 micromachines-13-01848-t004:** The highest temperatures on the wall with heat source for various inlet velocities and heat fluxes.

*v*_in_ (m/s)	0	0.0001	0.001	0.01	0.1	1
*T*_max_ for *q* = 10^6^ W/m^2^ (K)	/	368	359	350	336	312
*T*_max_ for *q* = 5 × 10^5^ W/m^2^ (K)	370	357	352	339	313	304
*T*_max_ for *q* = 2 × 10^5^ W/m^2^ (K)	364	352	338	316	304	302

## Data Availability

Not applicable.

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
