# Peer review of "Numerical Investigation on the Heat and Mass Transfer in Microchannel with Discrete Heat Sources Considering the Soret and Dufour Effects"

_micromachines, 2022, doi:10.3390/mi13111848_

Round 1
Reviewer 1 Report
This is an interesting piece of work. The following major points must be addressed carefully by revising the manuscript.
1. The introduction section should be written for the purpose of research and writing the paper.
2. Future work should be included in the conclusion section.
3. As, always the authors are suggested to read the paper for any typos or grammatical errors they may modify before final acceptance.
4. Authors should improve the abstract and objectives.
5. Add the novelty at the end of the introduction section.
6. Referenced all the key equations of the study.
7. Add nomenclature with SI units
8. Why the authors solved the dimensional equations instead of non-dimensional equations?
9. Some references in this field can enrich the introduction part such as:
· https://doi.org/10.1080/17455030.2022.2067371
· https://doi.org/10.3390/e20090668
Recommendation: I would recommend that authors do a thorough revision considering comments prior to a re-submission before this paper can be considered for publication in Micromachines. I would be happy to review a revised version.
Author Response
Response to Reviewer 1
This is an interesting piece of work. The following major points must be addressed carefully by revising the manuscript.
Thanks for your positive comment.
- The introduction section should be written for the purpose of research and writing the paper.
Response: Thanks for your constructive suggestion. We revised according to your comments.
- Future work should be included in the conclusion section.
Response: Thanks. We add the description of future work in conclusion section.
- As, always the authors are suggested to read the paper for any typos or grammatical errors they may modify before final acceptance.
Response: We check carefully the whole manuscript and revised.
- Authors should improve the abstract and objectives.
Response: Thanks, we modified the abstract and objectives.
- Add the novelty at the end of the introduction section.
Response: We add the innovation point at the end of introduction.
- Referenced all the key equations of the study.
Response: Thanks, we revised it according to your suggestion.
- Add nomenclature with SI units
Response: We add the nomenclature with SI units.
- Why the authors solved the dimensional equations instead of non-dimensional equations?
Response: The main reason is to highlight the present study is focused on microflows.
- Some references in this field can enrich the introduction part such as:
https://doi.org/10.1080/17455030.2022.2067371
https://doi.org/10.3390/e20090668
Response:These are cited in revised manuscript.
Author Response
Response to reviewer 2
The paper could be published after the following major revisions.
Thanks for your positive comment.
Comments/questions:
- Eq (5) should have a reference.
Response:Thanks, a reference for Eq 5) was added.
- The used software is not stated here. In the case that authors used a code for simulations,
the details of the discretization method and programming language should be added to the
paper.
Response: Thank you for your advice. We have added some descriptions on software and programming. We use Fluent 17.0, the discretization method can refer to the “HELP” in Fluent 17.0. Some factors, e.g. Soret and Dufour effects, can be taken into account by UDFs (User Defined Functions) programmed in C.
- 3 has no legend.
Response:We revised.
- The mesh independency report should be added to the paper before validation.
Response:We added the mesh independency report in the revised manuscript.
- How authors employed the Soret and Dufour effects separately in software simulation.
Response:We set a switch to call the UDFs for Soret or Dufour effects.
- In the introduction, different techniques such as using hydrophobic and hydrophilic
surfaces as well as numerical approaches for the simulation of fluid flow and heat transfer
could be discussed. In this regard, the following papers could be cited and discussed there.
https://doi.org/10.1140/epjp/s13360-019-00095-y
https://doi.org/10.5755/j01.mech.23.6.15804
Response: Above references are cited.
- An important question that comes from your contours. How author ensure that the flow is
single and the boiling doesn’t occur by applying heat flux at the heat sources? (Hint: in
most cases temperature is 374 K which is higher than the boiling point of water).
Response:Very good question! We did encounter this problem in the course of our calculations. We set a threshold value in the iteration and the iteration is interrupted when the fluid temperature exceeds 374 K.
Minor/typos:
1) In Eq. (2), andare vectors and should be changed to this form.
2) Eq. (1) should be corrected.
3) There is space between the number and unit e.g., 350K is not correct and should be 350 K.
4) Both sides of the “=” sign should be space
Response:We carefully checked the whole manuscript and revised!
Reviewer 3 Report
1. The abstract needs parameters and What are the main findings? Add to abstract by given numbers or percentage etc.
2. Present a more focussed survey on the specified topic. Also, at the end of the Introduction, clarify the novelty and gaps to be filled in the literature by the present attempt. Consider the following articles. “Convective flow of second grade fluid over a curved stretching sheet with Dufour and Soret effects” “Soret and Dufour effects on Oldroyd-B fluid flow under the influences of convective boundary condition with Stefan blowing effect”
3. The authors need to explain that the numerical approach used in the research is one of the appropriate solutions in the context of the research problem. What are the achievements of previous studies based on a numerical basis? Also, describe what has not been achieved?
4. It is helpful to complete the description of how to collect data, data processing scenarios, and interpret the data collection.
5. The discussion seems inadequate, and this is too short for a reputable international journal. Many graphical presentations are similar; it is worth thinking about expressing with other graphics
6. CPU time should be mentioned.
7. What software is used for the simulations? Was the code for the implemented by the authors or a function already existing in the software was used? If the code for the numerical method was taken from another publication or is part of the software used, please cite the resource.
8. Use vector graphic images, and avoid serif fonts in figures (use sans-serif types).
9. The conclusion must answer whether the proposed method can solve the research problem and achieve the objective. How can the numerical approach answer the existing issues? What is the most important result? What are the implications for science and technology development?
10. The writing of some references needs to be rechecked for accuracy
11. The conditions and assumptions are not clear. The main figure is not so clear update with all the assumption.
Author Response
Response to reviewer 3
- The abstract needs parameters and What are the main findings? Add to abstract by given numbers or percentage etc.
Response:We thank you for your constructive comments, upon which we have revised.
- Present a more focussed survey on the specified topic. Also, at the end of the Introduction, clarify the novelty and gaps to be filled in the literature by the present attempt. Consider the following articles. “Convective flow of second grade fluid over a curved stretching sheet with Dufour and Soret effects” “Soret and Dufour effects on Oldroyd-B fluid flow under the influences of convective boundary condition with Stefan blowing effect”
Response: Two references are added.
- The authors need to explain that the numerical approach used in the research is one of the appropriate solutions in the context of the research problem. What are the achievements of previous studies based on a numerical basis? Also, describe what has not been achieved?
Response: Thanks! On the basis of heterogeneous two-phase model (i.e. Buongiorno model), a modified model was obtained by adding a term for Soret effect in N-S equation, another term for Dufour effect in energy equation. Such models are suitable to model HMT in MCHS or heat exchangers with two species or discrete heat sources. We found that the pair-vortex due to the discrete heat sources can promote the HMT for natural convection or forced convection at lower velocity.
- It is helpful to complete the description of how to collect data, data processing scenarios, and interpret the data collection.
Response: Thanks for your suggestions. We revised according to the comment.
- The discussion seems inadequate, and this is too short for a reputable international journal. Many graphical presentations are similar; it is worth thinking about expressing with other graphics
Response: Thanks, we deleted some similar pictures and added some discussions.
- CPU time should be mentioned.
Response: Okay, we state the CPU times in revised manuscript.
- What software is used for the simulations? Was the code for the implemented by the authors or a function already existing in the software was used? If the code for the numerical method was taken from another publication or is part of the software used, please cite the resource.
Response: Thank you for your advice. We have added some descriptions on software and programming. We use Fluent 17.0, the discretization method can refer to the “HELP” in Fluent 17.0. Some factors, e.g. Soret and Dufour effects, can be taken into account by UDFs (User Defined Functions) programmed in C.
- Use vector graphic images, and avoid serif fonts in figures (use sans-serif types).
Response: Revised!
- The conclusion must answer whether the proposed method can solve the research problem and achieve the objective. How can the numerical approach answer the existing issues? What is the most important result? What are the implications for science and technology development?
Response: Thanks for your constructive suggestion. We revised according to your comments.
- The writing of some references needs to be rechecked for accuracy
Response: Thanks, we carefully checked all the references.
- The conditions and assumptions are not clear. The main figure is not so clear update with all the assumption.
Response: Thanks, we added some description on conditions and assumptions.
Round 2
Reviewer 1 Report
Now, the paper can be accepted for publication.
Reviewer 3 Report
The author has made edits to the work in response to the reviewer's comments. It is my opinion that the manuscript can be published in its current form.